# 3D Spheroid Human Dermal Papilla Cell as an Effective Model for the Screening of Hair Growth Promoting Compounds: Examples of Minoxidil and 3,4,5-Tri-O-caffeoylquinic acid (TCQA)

**DOI:** 10.3390/cells11132093

**Published:** 2022-06-30

**Authors:** Meriem Bejaoui, Aprill Kee Oliva, May Sin Ke, Farhana Ferdousi, Hiroko Isoda

**Affiliations:** 1Alliance for Research on the Mediterranean and North Africa (ARENA), University of Tsukuba, Tsukuba 305-8572, Japan; meriem.bejaoui@aist.go.jp (M.B.); aprillkeeoliva@gmail.com (A.K.O.); ferdousi.farhana.fn@u.tsukuba.ac.jp (F.F.); 2Research and & Development Center for Tailor-Made QOL Program, University of Tsukuba, Tsukuba 305-8572, Japan; 3AIST-University of Tsukuba Open Innovation Laboratory for Food and Medicinal Resource Engineering (FoodMed-OIL), AIST, University of Tsukuba, Tsukuba 305-8572, Japan; 4College of Agro-Biological Resource Sciences, School of Life and Environmental Sciences, University of Tsukuba, Tsukuba 305-8572, Japan; maysin.lucky.day@gmail.com; 5Faculty of Life and Environmental Sciences, University of Tsukuba, Tsukuba 305-8572, Japan

**Keywords:** 3D spheroid cell culture, human dermal papilla cells, minoxidil, TCQA, ECM

## Abstract

Dermal papilla cells (DPCs) are an important element of the hair follicle (HF) niche, widely used as an in vitro model to study hair growth-related research. These cells are usually grown in 2D culture, but this system did not show efficient therapeutic effects on HF regeneration and growth, and key differences were observed between cell activity in vitro and in vivo. Recent studies have showed that DPCs grown in 3D hanging spheroids are more morphologically akin to an intact DP microenvironment. In this current study, global gene molecular analysis showed that the 3D model highly affected cell adhesion molecules and hair growth-related pathways. Furthermore, we compared the expression of signalling molecules and metabolism-associated proteins of DPCs treated with minoxidil (an FDA-approved drug for hair loss treatment) and 3,4,5-tri-*O*-caffeoylquinic acid (TCQA) (recently found to induce hair growth in vitro and in vivo) in 3D spheroid hanging drops and a 2D monolayer using DNA microarray analysis. Further validations by determining the gene and protein expressions of key signature molecules showed the suitability of this 3D system for enhancing the DPC activity of the hair growth-promoting agents minoxidil and TCQA.

## 1. Introduction

It has long been known that the hair follicle (HF) undergoes continuous cycling that is categorized into three stages: expansion (anagen), apoptosis-driven regression (catagen), and inactivity (telogen) [1]. The HF contains different cell populations distinct in their function, molecular mechanism, and origin that throughout adult life are revived from niche-resided stem cells located in the bulge [2]. The growth cycle and cell regeneration are mainly controlled by a group of specialised fibroblastic cells of mesenchymal origin, known as the dermal papilla cells (DPCs) [3]. 

The longest stage within the hair cycle in humans is the anagen stage, which takes two to five years. The second phase lasts from a few days to a few weeks, and the last phase, telogen, usually takes three months in the normal adult hair cycle [1]. However, some disturbances can interfere with this process, causing premature termination of anagen and the early onset of the catagen and telogen phases [4]. These dysregulations may lead to the miniaturization of the hair shaft (HS) and/or a higher number of hairs in the catagen–telogen phase, which will result in hair loss and eventually alopecia [5,6,7]. These irregularities can appear in the DPCs, and therefore, discovering new insights regarding how to battle hair loss using cultured DPCs is still a focal point for many researchers. DPCs are mainly used in two-dimensional (2D) culture, and this system has been broadly studied as a form of hair growth in in vitro models for its simplicity and efficient culturing workflow [8]. However, 2D cultured DPCs have demonstrated no therapeutic effect on HF growth, and this raised questions as to the suitability of this model for studying hair growth cycle and hair regeneration [9]. The lack of regeneration activity in the 2D model can be explained by the necessity of communication between the epidermal cells and the DPCs, to regulate the hair growth cycle: for this reason, the cells primarily exist in the topographically complex three-dimensional (3D) extracellular environment within the HF [10]. Moreover, 2D cultured DPCs are usually attached to a collagen-coated polystyrene surface, and this configuration deeply constrains the actual behaviour of the cells [11,12]. Thus, 2D monolayer cell culture does not reflect the actual microenvironment that is composed of a specific cellular organization where the communication occurs from cell to cell or from cell to extracellular matrix (ECM) to assure the HF growth and regeneration [13].

Recently, an expansion of evidence and documentation has indicated that 3D cell culture systems more accurately represent the microenvironment surrounding the tissues [14]. In this system, the cells present morphological and physiological characteristics reflecting in vivo cellular responses, in contrast with 2D cultured cells [15,16]. Therefore, the 3D culture system is gaining attention not only because it reflects the dimensional organization of the cell, allowing the communication with neighbouring cells, but also it influences signal transduction, gene expression, and cellular behaviour [17,18]. Based on previous studies, 3D DPCs cultured in a spheroid system using the hanging drop technique may partially restore the hair inductivity potential of intact DPCs lost in 2D culture as they encourage the formation of a condensate-like structure [9,19]. However, other studies showed that the spheroid cell culture system may present limitations, specifically a somewhat weak interaction with the extracellular environment and an inconsistency in the expression of selected 3D markers [8,9].

Alopecia is an exceedingly prevalent hair loss condition affecting many people worldwide, and hence, it attracts demands for treatment [20]. Only two drugs have been approved by the Food and Drug Administration (FDA) as treatments for hair loss, one of which is minoxidil. However, various treatments have been proposed to manage this condition, such as with food supplements like curcumin and innovative treatments such as blue light [21,22]. In this context, our group previously published that the polyphenolic compound 3,4,5-tri-*O*-caffeoylquinic acid (TCQA) significantly activated Wnt/β-catenin signalling in vivo and in DPC for the promotion of the hair growth/pigmentation cycle [23,24]. Next, TCQA was found to promote the differentiation of 3D cultured human amnion epithelial stem cells while increasing ECM markers [25]. Consequently, there has been a high demand for using target compounds known to promote hair growth and ECM-related markers to overcome the limitations of 3D spheroid DPC culture. 

Hence in this paper, we first compared the expression of signalling molecules and markers of human DPCs in untreated 3D and 2D cell culture systems using microarray analysis. Our data demonstrated that cells grown in a 3D model can influence not only hair growth-related pathways but also immune system-related pathways, in contrast with the 2D cell monolayer. This study aims to use the hair loss approved drug minoxidil and the hair growth-promoting agent TCQA to further enhance and validate the 3D properties of DPC hanging drops by determining the molecular mechanism involved. 

## 2. Materials and Methods

### 2.1. Samples Preparation

Minoxidil was bought from Tokyo Chemical Industry (Tokyo, Japan). TCQA was synthesized with 97% purity. The two samples were prepared in 70% ethanol prior to dilution in papilla growth medium (as described below) for in vitro assay. 

### 2.2. Cells and Cell Culture 

Human DPCs were purchased from Cell Application Inc. (Tokyo, Japan). These cells are primary and derived from the temporal scalp of a 50-year-old healthy Caucasian male (Catalog No 602-05a, Lot No 1507). The cells can be maintained for two to three passages without losing their morphological characteristics with a doubling time of 30 h and a viability of 95%. Cells were kept and cultured using Papilla Growth Medium (Toyobo, Osaka, Japan) supplemented with growth factors: foetal calf serum (FBS), insulin transferrin triiodothyronine, bovine pituitary extract, and cyproterone solution (Toyobo, Osaka, Japan). The cells were constantly monitored for media change every two days and incubated under sterile conditions at 37 °C injected with 5% CO_2_ until they reached 80% confluency. Once confluent, cells were passaged according to the culture guidelines of Cell Applications Inc. and were seeded into a new coated flask with 10 mL collagen solution (Toyobo, Osaka, Japan). 

### 2.3. Establishment and Maintenance of 3D Culture

DPCs were cultured in a 3D spheroid hanging drop system [9]. Firstly, the cultured cells were washed with PBS and trypsinized, and 5 mL of the culture medium was added prior to centrifugation for 5 min at 200× *g*. Then cells were resuspended at 300 cells per µL. The 3D culture was established using the hanging drops method on the lid of a 100-mm petri dish, where a 10 µL hanging drop contains approximately 3000 cells. PBS was added inside the petri dish to completely cover the base of the dish. Three groups were then established, the control group (untreated cells) and groups treated with 0.1 and 10 µM of minoxidil and TCQA, respectively. The formation of spheroids using the hanging drops method was established after 48 h, and these cells were then used in the following experiments.

### 2.4. RNA Extraction 

The RNA was extracted from DPCs cultured in the 3D and 2D systems. For the 2D system, cells were seeded overnight (5 × 10^5^ cells per 100-mm dish), and then the papilla medium was renewed with the treatment (0.1 µM minoxidil and 10 µM TCQA). After 48 h, the medium was discarded, and a washing with cold PBS was followed by an Isogen extraction (Nippon Gene, Tokyo, Japan) according to the manufacturer’s directions. For the 3D hanging drops culture system (all three groups), the cells were first collected and centrifuged, and then the RNA was extracted as described above. For both systems, the RNA concentration was assessed using a NanoDrop 2000 spectrophotometer and used for the following experiments.

### 2.5. DNA Microarray Analysis

The RNA samples extracted from the 2D and 3D systems were prepared for microarray analysis using GeneChip^®^ 3′ Expression Arrays and 3′ IVT PLUS Reagent Kit (Affymetrix Inc., Santa Clara, CA, USA). In brief, (250 ng) was used to generate amplified and biotinylated aRNA following the manufacturer protocol. The array strips (HG-U219) were hybridized for 16 h in a 45 °C station, then washed, stained, and imaged in the GeneAtlas Fluidics and Imaging Station. 

For the analysis, firstly, Expression Console Software was used for the raw data normalization, allowing for the selection of genes with a *p*-value ≤ 0.05 (one-way between-subject ANOVA) to be designed as differentially expressed genes (DEGs). Further analysis was carried out using Transcriptome Analysis Console (TAC) version 4, Database for Annotation, Visualization, and Integrated Discovery (DAVID), Gene Set Enrichment Analysis (GSEA), ExAtlas: gene expression, and Meta-analysis [26,27,28]. 

### 2.6. Quantitative Real-Time PCR Analysis

The RT-PCR analysis was performed using a Superscript IV reverse transcription kit (Invitrogen, CA, USA). The cycling protocol was as follows: 95 °C for 10 min and 40 cycles at 95 °C for 15 s and at 60 °C for 1 min. TaqMan Universal PCR mix and TaqMan probes specific), *CTNNB1*, *ALPL*, NCAM, VCAN, RBbj, IL6, and TNF-α (Applied Biosystems, California, CA, USA), were used in the 7500 Fast Real-Time PCR Software 1.3.1. *GAPDH* (Hs 02786624_ g1) (Applied Biosystems, CA, USA) was used as an endogenous control. In order to calculate the relative mRNA expression levels using the endogenous control, the 2−ΔΔCt method was used.

### 2.7. Western Blot

The proteins were extracted from DPCs cultured in both the 2D and 3D systems and treated with and without 0.1 µM minoxidil for 48 h. Total protein extraction was achieved using radio-immunoprecipitation assay (RIPA) buffer (SIGMA, Saint Louis, MO, USA) and protease inhibitor and quantified with the 2-D Quant kit (GE Healthcare, Chicago, IL, USA).

Next, the proteins were separated in SDS-PAGE, and the transfer was carried out in a PVDF membrane (Millipore, Missouri, NJ, USA). After blocking, the membranes were incubated with primary antibodies against VCAN (Abcam, Rockford, IL, USA), NCAM (Abcam, Rockford, IL, USA), β-catenin 71-2700 (Thermo Fisher Scientific, Waltham, MA, USA), ALP (Abcam, Rockford, IL, USA), FGF1 (Abcam, Rockford, IL, USA), and GAPDH sc32233 (Santa Cruz 1 Biotechnology, Dallas, TX, USA) as endogenous controls. After overnight incubation at 4 °C, the secondary antibodies, goat anti-rabbit IRDye 800 CW or IRDye 680 LT goat anti-mouse, were added. Then, the signal was detected using a LI-COR Odyssey Infrared Imaging System (LI-COR, Lincoln, NE, USA).

### 2.8. Statistical Analysis 

The results were expressed as mean  ±  standard deviation (SD). Statistical analysis was performed using Student’s t-test when comparing two value sets. For microarray analysis, DEGs were selected based on ANOVA *p*-value < 0.05. Significantly enriched gene ontology terms were identified based on Modified Fisher Exact *p*-value for DAVID analysis and hypergeometric distribution of overlapping genes for GSEA analysis.

## 3. Results

### 3.1. Gene Profiling Analysis of HFDPCs Cultured in 3D Hanging Drops Compared with 2D Culture

Firstly, we performed DNA microarray analysis to determine the gene signature differences between the two studied systems. To understand the integrative response of 3D culture compared with the 2D, we assigned the cut-offs for the DEGs to be >2.0 or <2.0 (linear fold change). We found 1933 genes between the 2D and 3D samples to be consistently differentially expressed. Of these genes, 823 genes were upregulated, and 1110 genes were downregulated, as represented in the volcano plot that shows the distribution of the fold changes and the significance levels of the DEGs (Figure 1A).

The gene set enrichment analysis of the 3D control compared with the 2D control showed the upregulation of several previously defined hallmark gene sets associated with collagen regulation (GO:0005587, GO:0005518, GO:0062023), cell adhesion (GO:0007155), endothelial cell proliferation (GO: 0001938), and cellular response to UV (GO:0034644) (Figure 1B). Interestingly, immune system-significant genes were regulated in the 3D system but not in 2D (Figure 1C).

Next, we investigated the expression of two key adhesion molecules involved in ECM interaction, versican (VCAN) and the neural cell adhesion molecule (NCAM). Figure 1D,E displayed the upregulation of the expression of these molecules in the genes and proteins in DPCs cultured in 3D compared with the 2D method after 48 h treatment. Moreover, the gene expressions of CORIN, an anagen marker, and RBbj, a Notch pathway effector were investigated, and the results showed that 3D cells displayed a greater fold change compared with the 2D cells for both markers (Figure 1F). On the other hand, the expressions of TNFα and IL6, involved in inflammatory reaction, were reduced in the 3D spheroid system (Figure 1G). To summarize these findings, the highly regulated genes in 3D spheroid were classified in Table 1. These results showed that the 3D system enhances the cell–cell and cell–ECM interactions and hair growth-associated genes and confers on DPCs a better defence response and a more resistant immune system. 

### 3.2. Comparison of the DEGs in HFDPCs Treated with Minoxidil and TCQA in Both Systems 

This current study aims to check the effects of hair growth-promoting agents minoxidil and TCQA on a 3D spheroid system and then compare them with their effects on a 2D monolayer. For this purpose, the cells were cultured in 3D (Figure 2A) and 2D systems and treated with both samples at concentrations of 0.1µM and 10 µM for minoxidil and TCQA, respectively, for 48 h, and then global gene analysis was performed. We designed the control group as the untreated cells and the DEGs were to be >1.3 or <−1.3 (linear fold change). Firstly, we focused on the general effects of these two compounds on the 3D spheroid DPCs as illustrated in Figure 1 (compound vs. control). The scatter plots showed that in the case of 3D minoxidil, 5160 genes are upregulated (9.3% of total DEGs) and 6316 genes (7.72% of total DEGs) are downregulated, while for TCQA, 2450 (9.47% of total DEGs) are upregulated and 3575 (8.43% of total DEG) are downregulated (Figure 2B,C). PCA and clustering analysis demonstrated that the 3D cultured spheroids treated with minoxidil and TCQA exhibited a rather similar distribution to that of the DEGs compared with the control (Figure 2D,E).

Furthermore, a comparison between 2D and 3D systems upon minoxidil treatment (0 and 0.1 µM) was conducted where the top 100 DEGs were subjected to hierarchical clustering. These results demonstrated that the top genes displayed different expressions between 3D minoxidil and 3D control and 2D minoxidil and 2D control (Appendix A).

### 3.3. Comparison of DEG between 3D and 2D System upon Minoxidil Treatment 

First, to know minoxidil’s mechanism of action, we analysed the gene profiling of 2D minoxidil vs. 2D control and 3D minoxidil vs. 3D control. The results revealed that 2D minoxidil affected CTNNB1, Wntless (WLS), fibroblast growth factor 7 (FGF7), bone morphogenetic protein 1 (BMP1), and collagen 5A (COL5A), genes linked with the Wnt/β-catenin, FGF, BMP, and collagen pathways (Appendix A). The downregulation of genes linked with collagen breakdown including the matrix metalloproteinase molecules MMP24, MMP7, and MMP8 and Wnt/β-catenin repression such us TAXIBP3 was observed (Appendix A). These data showed that minoxidil treatment in the 2D-cultured HFDPCs enhanced hair growth-associated genes, mainly Wnt/β-catenin pathway. 

On the other hand, minoxidil treatment in the 3D-cultured DPCs upregulated CTNNB1, ABCC11 (ATP binding), collagen adhesion, and cell adhesion molecules like NCAM and VCAN (Table 2). Moreover, the expression of genes involved in telogen maintenance and stem cell quiescence was significantly decreased upon minoxidil treatment in 3D-cultured DPCs (Table 3). 

In addition, in the heat map created to compare cells treated with minoxidil in the 3D and 2D culture systems, we observed the enhancement of the expression of genes significant for cell–cell adhesion (GO: 0098609), Wnt signalling (GO: 0060071), positive regulation of canonical Wnt signalling (GO: 0090263), tricarboxylic acid cycle (GO: 0006099), and cell cycle arrest (GO: 0007050). On the other hand, 3D minoxidil treatment inhibited genes linked with interferon production (GO: 0032728), NF-kappaB signalling (GO: 0043123), and apoptotic signalling (GO: 1900740) (Appendix A). The molecular function and the Kegg pathway analysis of the DEGs showed that minoxidil treatment in the 3D spheroid DPCs upregulated ATP binding pathways, cell cycle, and fatty acid metabolism, in contrast with 2D minoxidil treatment (Appendix A). These current data demonstrated that minoxidil stimulated hair growth-associated genes, ATP, and collagen binding, and the observed effect was significantly higher in the 3D culture system. Moreover, to prove that the upregulation of hair growth-, cell adhesion-, and collagen binding-associated genes was due to minoxidil treatment and not only to the 3D structure, the DEGs between the 3D control and the 2D control and 3D minoxidil and 2D minoxidil were compared. Appendix A illustrated the obtained data, and the common upregulated genes between these two sets are only 12; additionally, the 3D minoxidil upregulated genes were more linked with cell cycle arrest and inflammation response compared with the 2D minoxidil. For the downregulated genes between the previously stated sets, the common genes are 11, and 3D minoxidil significantly decreased the expression of genes linked with inflammation and cell division (Appendix A). 

These results showed that minoxidil treatment on 3D cultured DPCs highly affected hair growth-associated pathways and markers and cell cycle arrest signalling while conferring an anti-inflammatory activity.

### 3.4. 3D Cultured DPCs Treated with Minoxidil Enhanced the Gene and Protein Expression of Hair Growth Markers 

Firstly, the gene expressions of NCAM and VCAN were assessed upon minoxidil treatment in the 3D spheroids, and the results revealed that minoxidil enhanced the expression of these markers, confirming that the treatment did not affect the establishment of the 3D structure (Figure 3A). 

The highly affected pathways by minoxidil treatment whether in 2D or 3D culture system are the Wnt/β-catenin and FGF pathways, as well as dermal papilla marker alkaline phosphatase or ALP. The gene and protein expressions of these markers were checked, and the results showed a positive stimulation upon treatment. We take the example of β-catenin, where minoxidil treatment enhanced the gene expression up to 1.8- and 2.8-fold and the protein expression up to 1.4- and 2.2-fold, respectively, in the case of the 2D and 3D structures (Figure 3B,C). We observed the same tendency in the case of FGF1 and ALP. These data demonstrated that minoxidil treatment in the 3D system had a higher effect on the regulation of hair growth markers in DPCs compared with the 2D model.

### 3.5. Characteristics of 2D and 3D Cultured HFDPCs Treated with TCQA

In this study, we used another compound to test the efficacy of the 3D system, TCQA, which was previously reported to stimulate the anagen phase and the hair growth cycle in C3H mice and in HFDPCs [24]. 

To elucidate the characteristics of the cells cultured in 2D and 3D and simultaneously treated with TCQA, gene profiling analysis was conducted. The result showed that the gene sets for the Wnt signalling pathway, epithelial–mesenchymal cell differentiation, collagen, cell–cell adhesion, and fibroblast proliferation were significantly upregulated upon treatment (Figure 4A,B). To further determine the extent of the gene expression differences between the two cell culture methods used, the top upregulated and downregulated genes were identified and classified in tables. The specific Wnt signalling and anagen induction genes that were modulated were CTNNB1, Wnt11, WLS, and ALP, along with FGF1 and FGF5 (Appendix A). Other factors involved in HF regeneration and hair cycle regulation such as SOX4, NRAS, BMPR-1A, and BMP5 were enhanced upon treatment as well. It is worth mentioning that VCAN, a marker of 3D spheroid formation and fibroblast differentiation, was upregulated in both the 2D and 3D systems. While this is true for both 2D and 3D cultures, the genes in the TCQA-treated 3D culture exhibited a more than double fold change, in contrast with the fold change in the 2D monolayer cultured cells (Appendix A). CTNNB1 expression was stimulated up to 1.62-fold in the TCQA-treated 2D HFDPCs, while in the TCQA-treated 3D HFDPCs, the expression was up to a 43.57-fold change. These results showed that the effect of TCQA on stimulating hair growth-related genes was more pronounced and significantly higher in the 3D spheroid system compared with the 2D. Appendix A illustrated the repression of genes linked with Wnt/β-catenin inhibition, telogen regulation, anagen delay, and collagen breakdown for both the 2D and 3D cultures (TCF3, LDB3, AXIN2, GSK3B, and EGR1). We also noticed the pattern that the fold change was further decreased in the case of 3D culture compared with the 2D upon TCQA treatment. In addition, the DEGs and their related functions between 3D TCQA and 3D control and between 2D TCQA and 2D control were compared, and Venn diagram was created for both upregulated and downregulated genes (Figure 4C,D). The results revealed that few genes were common between the two compared sets; the top DEGs were summarized in Appendix A.

Furthermore, the results of microarray in this study were compared with our previous in vivo microarray results from the back skins of eight-week-old C3H male mice treated topically for one month with TCQA upon shaving [24], as displayed in Table 4. We observed that 3D-cultured TCQA-treated DPCs exhibited an extraordinarily high level of fold change compared with 2D, and the in vivo data mainly linked to canonical Wnt signalling previously reported to be stimulated by TCQA (Table 4). 

Additionally, a heat map was created to compare the highly regulated functions including hair growth, pigmentation, keratin, and collagen between 2D control, 2D TCQA, and 3D TCQA (Figure 5A). Figure 5A also showed that 3D TCQA treatment had a higher effect compared with the 2D system and the untreated cells in all mentioned functions. Moreover, some of the DEGs were validated by RT-PCR, and results showed that the gene expression of the 3D culture structure markers NCAM and VCAN were enhanced upon TCQA treatment (Figure 5B). This suggests that TCQA’s clearly greater ability to express these genes in 3D is due to better cell–cell interaction between the dermal spheroids. CTNNB1, ALPL, and FGF1 are significantly expressed by TCQA in the genes of both the 2D- and 3D-treated cells; however, the comparison showed that the 3D cells treated with TCQA expressed the said genes significantly higher than 2D (Figure 5B). Thus, these data revealed that TCQA enhanced hair growth-related pathways and decreased telogen- and Wnt repression-significant genes, and this effect is more evident in 3D culture systems.

## 4. Discussion

HF neogenesis is considered an attractive target for regenerative medicine purposes and is of widespread clinical interest, in part due to the strong foundation of knowledge underpinning the HF biology [9]. One of the main approaches inducing the anagen phase of the HF cycle is the replenishment of DPCs using in vitro 2D-cultured systems; however, this technique was limited as the cells can lose their hair-inducing capacity over time because they are cultured on a flat, plastic surface [29]. Additionally, numerous studies have shown that 2D-cultured DPCs were unable to induce the anagen–telogen transition upon in situ implantations after six passages, as these cells need to agglomerate in the HF to be highly active [30,31,32]. To overcome these problems, 3D spheroid culture is considered to be an efficient tool for regaining DPCs’ inductive capacity, which may enable them to induce de novo HF in human skin provided there is the required cross-talk with the surrounding environment [5,33]. Nevertheless, extensive knowledge of the molecular mechanisms governing the regenerative process is necessary. Therefore, in this study, a global gene profiling analysis was performed to compare 3D spheroids with 2D-cultured DPCs to determine the signature molecules regulating the inductive phenotype of DPCs. In addition, we compared the effect of the alopecia-approved drug minoxidil and the polyphenolic compound TCQA, which was recently found to enhance hair growth in vivo and in vitro, in 3D and 2D culture systems.

Generally, the growth of DPCs within a spheroid structure facilitates cell aggregation and cohesion, and their loss may bring, papilla miniaturization, the disturbance of adult HF morphogenesis, and eventually hair loss [34,35]. Our data confirmed the previous finding that in the case of 3D spheroid DPCs, the GO of collagen regulation, cell adhesion, ECM, and cell proliferation were positively stimulated (Figure 1). The ECM plays a major role in cell growth and adhesion and the cell–cell interaction. A common physiological feature of DPCs is their close contact with the ECM, which is proved to be further enhanced in 3D spheroid culture [36,37]. For in vitro studies involving DPCs, NCAM and VCAN have been used as major indicators for the inductive capabilities of these cells, as decreases in their expression correlate with the decline of DPCs’ inductive capacities [38,39]. NCAM is involved in dermal condensation and hair induction, whereas VCAN is not only implicated in matrix assembly and structure and in cell adhesion but also in HF development and cycling [40,41,42]. Our study has shown that in 3D spheroids, the expression of these DPC markers was higher compared with the 2D culture system (Figure 1). Moreover, our data showed the upregulation of the hair growth markers RBbj and CORIN and the downregulation of the inflammatory effectors IL6 and TNFa (Figure 1). RBPj is an effector of Notch signalling, involved in the activation of the transcription of hair growth-associated genes. Meanwhile, CORIN is considered a DP marker upregulated during the anagen phase of the hair growth cycle [43,44]. On the other hand, TNF-α is reported to inhibit keratinocyte proliferation, leading to decreasing matrix cell size [45]. IL6 plays a role in HF regression by suppressing the elongation of the HS and decreasing the cell proliferation of the hair matrix [46]. 

All together, restoring human DPC inductivity via the artificial recreation of the 3D cell architecture using 3D spheroid cultures is considered to be an efficient method for generating hair regrowth [47]. Nevertheless, while this technique allowed for the partial restoration of DPCs’ native signature and transcriptome, it has shown a limitation on human–human recombinant assay as the 3D spheroid could only stimulate the hair induction by 15%, thus the need to address new strategies in order to properly recover the inductive capacities of DPCs [9]. Other studies have identified ECM analogues as a method for 3D culture system, where the cells are embedded in natural or synthesized scaffolds such as collagen type 1 and MatrigelTM; however, the compound’s biodegradability may compromise the maintenance of the biochemical properties [48]. Another alternative that was proposed to mimic the ECM environment is animal-derived polymers, but ethical and variability concerns were raised in this case [49]. For this purpose, hydrogels from a non-animal origin such as alginate are gaining enormous consideration due to their applicability for tissue regeneration such as improving the in vitro performance of prepubertal lamb oocytes [50,51]. Thus, in order to improve the efficacy of our 3D model, we have proposed using compounds and biomolecules that can activate hair growth- and ECM-related pathways in 3D spheroid systems to further enhance DPC inductive properties [52,53]. Therefore, in this current research, minoxidil, the FDA-approved drug to treat alopecia, and TCQA, a polyphenolic compound reported to stimulate hair growth, were tested in a 3D spheroid system. 

Minoxidil affects the hair growth cycle whether in animal models or in human clinical trials by shortening the length of the telogen phase and increasing the rate of DNA synthesis in the HF bulbs, causing early entry to the anagen phase [54,55,56]. An experimental study with 2D cultured DPCs has reported that minoxidil prolonged the anagen phase by inducing prostaglandin E2 and β-catenin activity and stimulating follicular proliferation and differentiation [57,58]. To the best of our knowledge, minoxidil’s mechanism of action in 3D spheroid DPCs is not yet explored. Here, we reported that minoxidil upregulated the GO of hair growth-, cell adhesion-, ATP binding-, and collagen-associated factors in a 3D system compared with 2D (Figure 2). The proliferation of DPCs during the anagen phase is usually followed by an increase in the ATP content, resulting in the expansion of cell size, which will directly affect the differentiation of keratinocytes into the HS [59,60]. The extracellular matrix category included several collagens in addition to other transcripts that are expressed in the intact DP [61], which were found to be more present in the 3D minoxidil-treated spheroids compared with the 2D ones (Figure 2, Table 2). Moreover, the gene expression analysis data were supported by the increased gene expressions of NCAM and VCAN and the increased gene and related protein expressions of the hair growth markers *ALPL* (ALP), *FGF1* (FGF1), and β-catenin (*CTNNB1*) in the 3D spheroids compared with the 2D upon minoxidil treatment (Figure 3). As mentioned earlier in this paper, several molecular markers have been used as markers of HF inductive properties, including VCAN and NCAM. Numerous studies have discussed that increasing VCAN expression in DPCs enables HF regeneration and stimulates growth cycle initiation [62,63]. ALP is a known marker of DPCs that is overexpressed during the growth phase of the hair cycle, while FGF1 is required in the promotion of the hair regrowth on the backs of C57BL/6 mice [7,64,65]. The role of β-catenin is well documented and established in the regulation of hair growth, as it is highly expressed during the anagen phase in DPCs and its absence induces a premature catagen phase [5,66,67].

In sum, the expression of hair growth markers was more pronounced in the 3D model compared with the 2D system, and minoxidil treatment further enhanced the efficacy of the 3D spheroid system in DPCs. These data are similar to what has been previously reported with other 3D methods such as scaffold and encapsulation where the DP marker ALP and the anagen-prominent genes β-catenin and FGF were highly regulated [68,69]. Next, 3D minoxidil downregulated genes, repressing the Wnt/β-catenin pathway (*AXIN2*) and telogen markers such as *NFATC1* (Table 3). Axin2 is involved in β-catenin phosphorylation, degradation, and non-translocation to the nucleus, leading to a shortened anagen phase in the HF. Nfatc1 plays a role in HF stem cell quiescence, which explains its high expression during the telogen phase, and then decreases during the transition to anagen, favouring stem cell differentiation [70,71,72].

On the other hand, TCQA was previously reported to have the ability to promote hsair growth in CH3 mice and in HFDPCs via the activation of β-catenin and its related genes [24]. Therefore, TCQA’s capability to activate Wnt/beta-catenin signalling, and the related hair growth pathways in 3D spheroid DPCs was investigated. TCQA-treated 3D culture exhibited a higher expression of gene sets related to Wnt/beta-catenin pathway, cell proliferation and collagen compared to the 2D culture (Figure 4 and Figure 5). The canonical Wnt pathway serves as the central regulator of hair morphogenesis and cycling, and different types of Wnt molecules have various roles in HF regeneration [73]. Additionally, 3D-TCQA-treated cells highly expressed some collagen-related genes like COL3A1 (Appendix A and heat map in Figure 5). This is a type III collagen, also known as fibrillar collagen found in the skin mainly associated with the papillary dermis, produced by intact DPCs, and responsible for HF development [74]. DPCs produce a wide diversity of ECM compounds, including collagens widely reported for HF growth and development. The changes in the composition and contents during the different phases of the hair growth cycle highlight the role of the ECM in regulating the different differentiation and proliferation processes. The expressions of the various ECM compounds are highly stimulated in the 3D culture system using a scaffold matrix or encapsulation, which correlates with our findings [53,75,76]. Additionally, the comparison of the genes between TCQA treated-DPCs in the 2D and 3D systems with TCQA-treated mice skin previously published [24], listed in Table 4, clearly showed that the fold changes in the selected genes greatly differ. The fold change in the TCQA-treated 3D culture is more than double for some genes compared with the 2D monolayer culture, and surprisingly, the fold change for the 3D system was higher than in the in vivo model. Taken altogether, despite TCQA’s positive effect on 2D-cultured DPCs, the full potential of the compound when studied in 2D culture was highly compromised. 

Thus, restoring cellular aggregation can mimic the actual DPC niche, and therefore, the effect of this compound was higher in 3D spheroid DPCs compared with the 2D system. 

Overall, we have demonstrated that the partial reprogramming of HFDPCs by spheroid formation with the addition of hair growth-stimulating compounds can restore the inductive capacities of DPCs. These findings represent a significant leap for both 3D spheroid DPC culture and minoxidil and TCQA as hair-promoting agents. Nevertheless, further studies should be pursued using 3D-treated spheroids in ex vivo or in co-culture systems.

## Figures and Tables

**Figure 1 cells-11-02093-f001:**
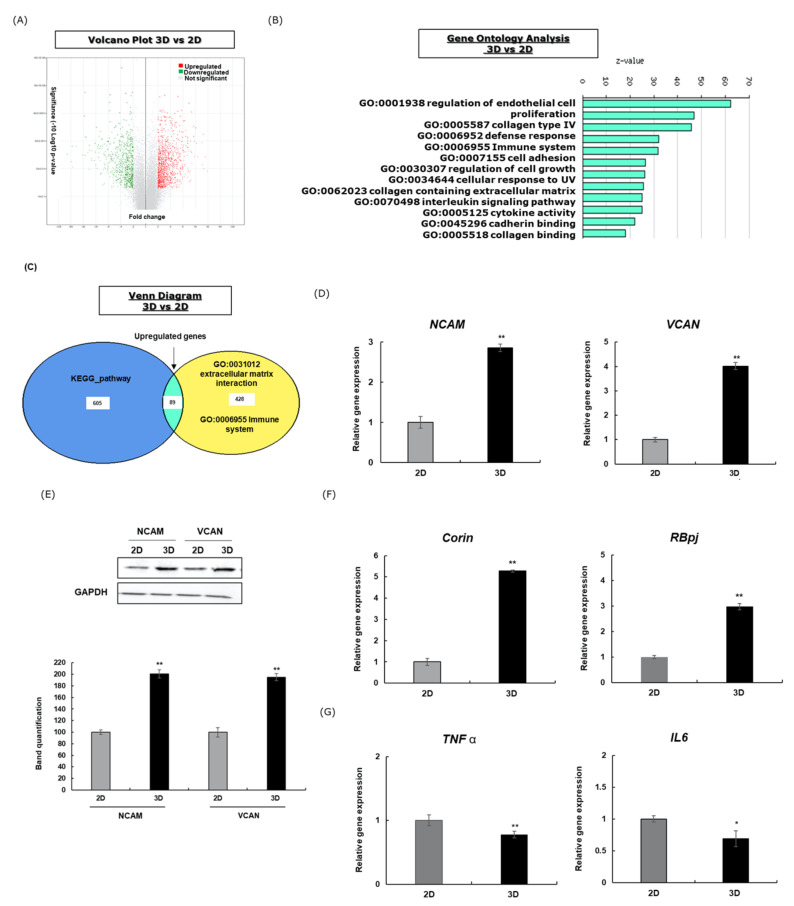
The gene profiling analysis of HFDPCs cultured in 3D spheroids vs. 2D monolayer. (**A**) Volcano plot showing the DEGs. Red, green, and grey illustrate the upregulated, downregulated, and nonsignificant DEGs, respectively. (**B**) The top 12 significantly enriched biological processes by the upregulated DEGs (analysed using Ex atlas). (**C**) Venn diagram showing common and unique sets of DEGs. (**D**) The gene expression of VCAN and NCAM after 48 h. (**E**) VCAN and NCAM protein expression after 48 h. Band intensities were assessed using LICOR. The results represent the mean ± SD of three independent experiments. * *p* ≤ 0.05, ** *p* ≤ 0.01. (**F**) The gene expression of DPC marker CORIN and, Notch pathway effector RBbj after 48 h. (**G**) Gene expression of inflammation associated genes after 48 h. The mRNA level was quantified using TaqMan real-time PCR.

**Figure 2 cells-11-02093-f002:**
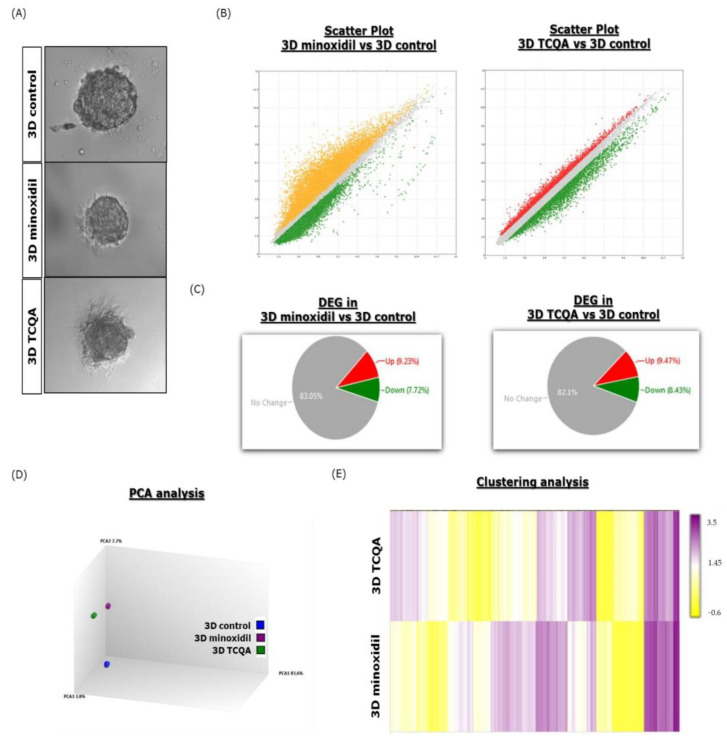
Microarray analysis of 3D HFDPCs treated with minoxidil and TCQA. (**A**) 3D spheroid HFDPCs treated with 0 (control), 0.1 µM minoxidil, and 10 µM TCQA for 48 h. (**B**) Scatter plot showing the DEGs. Yellow or red, green, and grey illustrate the upregulated, downregulated, and nonsignificant DEGs, respectively, of 3D treated with minoxidil and TCQA vs. 3D control. (**C**) The chart represents the DEG distribution, and the colours are assigned as above. (**D**) PCA analysis of 3D control, 3D minoxidil, and 3D TCQA. (**E**) Clustering analysis of 3D minoxidil and 3D TCQA.

**Figure 3 cells-11-02093-f003:**
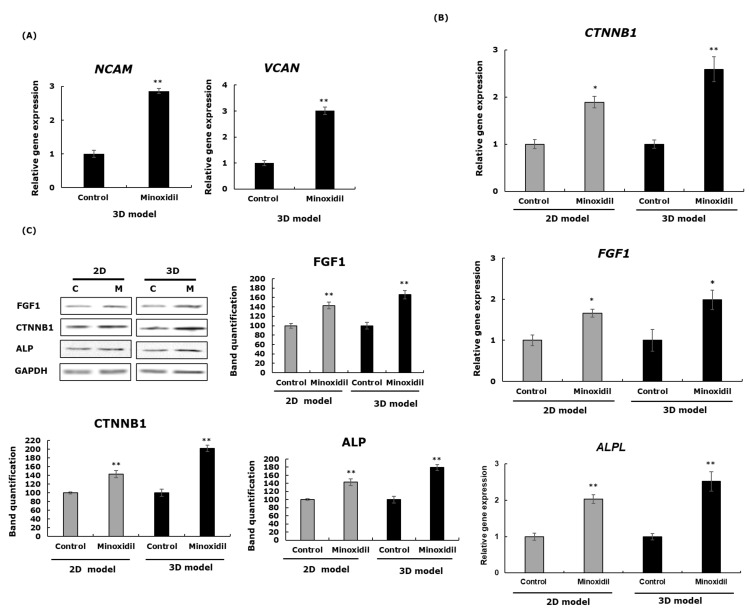
The expressions of ECM and hair growth markers upon minoxidil treatment in the 3D and 2D systems. (**A**) The gene expressions of the ECM markers VCAN and NCAM after 48 h. (**B**) The gene expressions of the hair growth markers CTNNB1, ALPL, and FGF1 after 48 h. For A and B, the mRNA level was quantified using TaqMan real-time PCR. The results represent the mean ± SD of three independent experiments. (**C**) β-catenin, ALP, and FGF1 protein expressions after 48 h. The band intensities were assessed using LICOR. The results represent the mean ± SD of three independent experiments. * *p* ≤ 0.05, ** *p* ≤ 0.01.

**Figure 4 cells-11-02093-f004:**
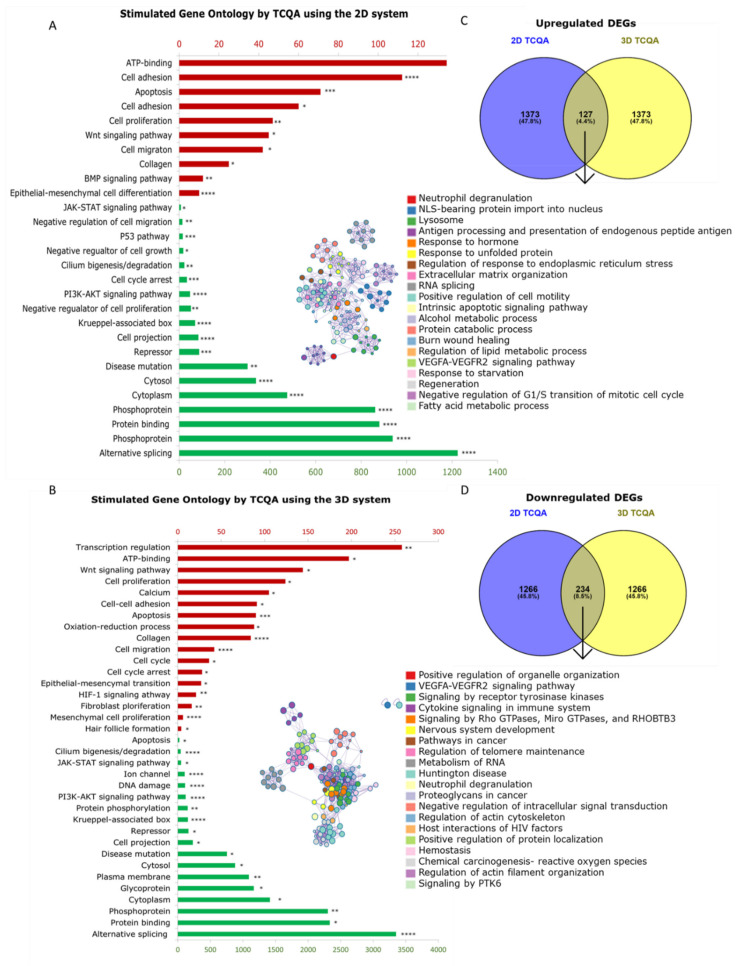
The significantly enriched DEG biological processes stimulated by TCQA-treated 2D- and 3D-cultured HFDPCs. (**A**) The top significantly enriched gene ontologies (GO) by the upregulated and downregulated DEGs in 2D-cultured HFDPCs treated with TCQA. (**B**) The top significantly enriched gene ontologies (GO) in the upregulated and downregulated DEGs of 3D-cultured HFDPCs treated with TCQA. The red bar plots represent the upregulated DEGs, while the green bar plots represent the downregulated DEGs. (**C**) Venn diagram showing the upregulated genes of TCQA-treated 2D- and 3D-cultured HFDPCs and the number of overlapping genes. The arrow points to the biological process obtained from the overlapping genes visualized using a cluster generated from Panther. (**D**) Venn diagram showing the downregulated genes of TCQA-treated 2D- and 3D-cultured HFDPCs and the number of overlapping genes. The arrow points to the biological process obtained from the overlapping genes visualized using a cluster generated from Panther Classification system (www.pantherdb.org). An ANOVA was used to assess the level of significance between the groups. * Statistically significant (*p*-value ≤ 0.05), ** statistically significant (*p*-value ≤ 0.01), *** statistically significant (*p*-value ≤ 0.001), **** statistically significant (*p*-value ≤ 0.0001). For GO analysis with DAVID, Modified Fisher Exact *p*-value was used.

**Figure 5 cells-11-02093-f005:**
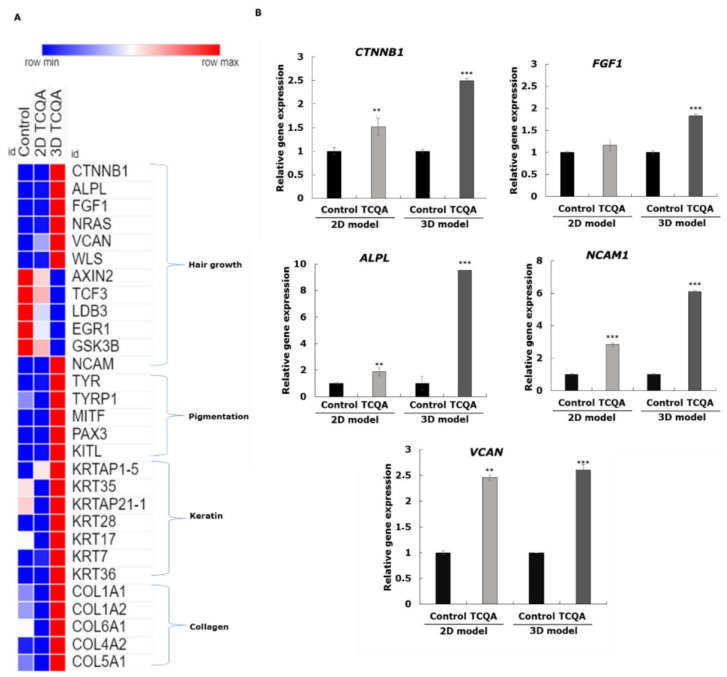
The enriched biological function categories regulated by the TCQA-treated 2D- and 3D-cultured HFDPCs and validation by gene expression analysis. (**A**) The heat map shows the comparison between 2D, 3D, and control. This displays genes related to several biological functions like hair growth, collagen, pigmentation, and collagen. The heat map was generated using the online tool Morpheus Broad Institute (https://software.broadinstitute.org/morpheus). (**B**) The gene expression of CTNNB*1*, *ALPL*, *FGF1*, NCAM1, and VCAN. The result represents the mean ± SD from three independent experiments. A T-test was used to assess the level of significance between the groups.), ** statistically significant (*p*-value ≤ 0.01), *** statistically significant (*p*-value ≤ 0.001).

**Table 1 cells-11-02093-t001:** DEGs in the 3D spheroid system compared with the 2D culture *.

Gene Symbol	Gene Name	Biological Function	Fold-Change	*p* Value **
*OXTR*	Oxytocin receptor	Cell-surface-receptor signalling pathway; oxidative stress	24.82	0.0002
*COL11A1*	Collagen alpha-1 (XI) chain	Extracellular matrix binding	21.67	0.003
*SSTR1*	Somatostatin receptor type 1	Expressed during anagen; HF immune system	21.03	0.001
*NEDD9*	Enhancer of filamentation 1	Cell adhesion; hair placode signature genes	18.48	0.004
*HAPLN1*	Hyaluronan and proteoglycan link protein 1	Cell adhesion; cell–cell communication	14.06	0.005
IL8	Interleukin-8	Immune response; inflammation	−179.99	0.004
*TNFAIP6*	Tumor necrosis factor-inducible gene 6 protein	TNF-α signalling; inflammation	−32	0.001
RANBP3L	Ran-binding protein 3-like	Termination of BMP signalling; inhibition of mesenchymal stem cell differentiation	−31.18	0.003
GDF15	Growth/differentiation factor 15	Stress response cytokine, expression increases with inflammation and injury	−25.96	0.045

* Gene functions were obtained from Mouse Genome Informatics (MGI). ** ANOVA was performed to assess the level of significance between groups. The gene expression was considered significant when the fold change was ≥2-fold (control vs. TCQA).

**Table 2 cells-11-02093-t002:** The top upregulated genes in the 3D minoxidil-treated cells (vs. 3D control) *.

Gene Symbol	Gene Name	Biological Function	Fold-Change	*p* Value **
*CTNNB1*	Catenin (cadherin associated protein), beta 1	Anagen regulation; hair follicle morphogenesis; positive regulation of fibroblast growth; dermal papilla cell proliferation	19.06	0.049
*TRH*	Pro-thyrotropin-releasing hormone	Hair shaft elongation; anagen prolongation; hair matrix keratinocyte proliferation	7.25	0.011
*RASGRF1*	Ras-specific guanine nucleotide-releasing factor	Hair follicle morphogenesis; ATP binding	6.75	0.020
*FERMT1*	Fermitin family homolog 1	Cell adhesion; keratinocyte proliferation and morphogenesis	6.03	0.018
*ABCC11*	ATP-binding cassette sub-family C member 11	ATP binding	5.88	0.029
*CXXC5*	CXXC-type zinc finger protein 5	Involved in MAPK pathway; cell cycle arrest	5.11	0.021
*NCAM*	Neural cell adhesion molecule	Hair morphogenesis; highly expressed during anagen; keratinocyte segregation and differentiation	4.72	0.085
*COL28A1*	Collagen alpha-1(XXVIII) chain	Hair shaft strength; matrix assembly	4.62	0.006
*KRT40*	Keratin, type I cytoskeletal 40	Late hair differentiation	4.06	0.010
*VCAN*	Versican	Collagen-containing extracellular matrix; highly expressed during anagen	3.23	0.008

* Genes functions were obtained from Mouse Genome Informatics (MGI). ** ANOVA was performed to assess the level of significance between groups. The gene expression was considered significant when the fold change was ≥2-fold (control vs. TCQA).

**Table 3 cells-11-02093-t003:** The top downregulated genes in the 3D minoxidil-treated cells (vs. 3D control) *.

Gene Symbol	Gene Name	Biological Function	Fold-Change	*p* Value **
*NFATC1*	Nuclear factor of activated T cells, cytoplasmic, calcineurin dependent 1	Stem cell quiescence	−8.22	0.001
*STEAP3*	STEAP family member 3	Inhibition of cell cycle arrest; downregulation of P53 pathway	−7.96	0.003
*LFNG*	LFNG O-fucosylpeptide 3-beta-N-acetylglucosaminyltransferase	Notch pathway inhibition; decreased hair development	−7.07	0.011
*PRUNE2*	Protein prune homolog 2	Regulation of tumour cell differentiation, survival, and aggressiveness	−6.82	0.032
*SKIV21*	Superkiller viralicidic activity 2-lik	ATP degradation	−5.45	0.007
*TRABD2B*	TraB domain containing 2B	Negative regulator of the Wnt pathway; Wnt protein cleavage	−5.04	0.027
*MAF*	Avian musculoaponeurotic fibrosarcoma oncogene homolog	Oncogene; embryonic marker of development	−4.78	0.062
*STYK1*	Serine/threonine/tyrosine kinase 1	Tumour cell invasion and metastasis	−3.42	0.012
*AXIN2*	Axin 2	β-Catenin phosphorylation and degradation	−3.33	0.028
*SIX3*	Sine oculis-related homeobox 3	Wnt/β-Catenin repression	−2.74	0.005

* Gene functions were obtained from Mouse Genome Informatics (MGI). ** ANOVA was performed to assess the level of significance between groups. The gene expression was considered significant when the fold change was ≥2-fold (control vs. TCQA).

**Table 4 cells-11-02093-t004:** Comparison between the TCQA-treated HFDPCs in 2D, in 3D, and in vivo [24] *.

Gene Symbol	Gene Name	Biological Function	Fold-Change		
			2D	3D	In Vivo
*CTNNB1*	Catenin (cadherin associated protein), beta 1	Anagen regulation; hair follicle morphogenesis; positive regulation of fibroblast growth; dermal papilla cell proliferation	1.62	43.57	3.2
*FGF1*	Fibroblast growth factor 1	Hair follicle morphogenesis; hair growth cycle regulation	1.53	26.45	3.22
*NRAS*	Neuroblastoma RAS viral oncogene homolog	Cell proliferation	1.53	16	1.2
*ALPL*	Alkaline phosphatase	Wnt/β-catenin pathway regulator	1.2	8	2.33
*BMPR-1A*	Bone morphogenetic protein	Skin development; hair follicle growth	1.14	64.89	1.7
*VCAN*	Versican	Cell aggregation marker; cell adhesion and proliferation	4.61	12.21	1.6
*WLS*	Wntless	Wnt secretion and pathway	1.6	16.38	3
*AXIN2*	Axis inhibition protein 2	Wnt-responsive gene	−1.5	−5	−2.82
*TCF3*	Transcription factor 3	Wnt/β-Catenin repression	−1.1	−5	−2.82
*LDB3*	LIM Domain Binding 3	Wnt-responsive gene; regulation of hair follicle during telogen	−1.14	−2.7	−8.7
*EGR1*	Early growth response 1	Negative regulation of Wnt/β-Catenin; upregulated with aging	−1.7	−4	−2.6
*GSK3B*	Glycogen synthase kinase 3 beta	Phosphorylation of β-catenin	−1.5	−6	−5

For all Tables * Gene functions were obtained from Mouse Genome Informatics (MGI). ANOVA was performed to assess the level of significance between groups. The gene expression was considered significant when the fold change was ≥2-fold (control vs. TCQA).

## Data Availability

I The data that support the findings of this study are available within the paper. The microarray data have been deposited in the NCBI, GEO database (accession: GSE178510 and GSE178637) https://www.ncbi.nlm.nih.gov/geo/query/acc.cgi?acc=GSE178510. https://www.ncbi.nlm.nih.gov/geo/query/acc.cgi?acc=GSE178637 (accessed on 24 June 2021).

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
