# Peer review of "3D Spheroid Human Dermal Papilla Cell as an Effective Model for the Screening of Hair Growth Promoting Compounds: Examples of Minoxidil and 3,4,5-Tri-O-caffeoylquinic acid (TCQA)"

_cells, 2022, doi:10.3390/cells11132093_

Round 1
Reviewer 1 Report
The paper already underwent multiple rounds of revisions and it is much more improved compared to the original submission. i think that this manuscript may be considered for publication in its current form
Author Response
We thank the reviewer for all his suggestions and comments that helped to further improve the paper.
Reviewer 2 Report
While the revised version of the manuscript comes with new figures and improved data presentation, the quality of the manuscript is still not suitable for publication. English grammar has still to be improved; I advise you to ask an English native speaker to review it. There are still numerous typos throughout the revised version of the manuscript, probably more than in the original version. In some cases, I want to believe that something went wrong with the PDF conversion (I have attached the version that I received for the authors to check) but many of them were already in the original version that I reviewed. Most of my previous comments were not addressed or not properly, although authors claim they did. Authors should still drastically improve their manuscript before it can be found suitable for publication.

Author Response
- We thank the reviewer for his suggestions. We think as well that the track changes were difficult to follow, for this reason, we have replaced it with a yellow highlight. We have further improved the English and the typos.
Regarding the previous comments, we have tried to answer them and follow the suggestions. But for some comments like 2, we couldn’t perform the experiment and we kindly tried to explain the reason. Please allow us to go through the old comments again and give more explanation.
Comment 1: In both 3D and 2D models, gene expression of CTNNB1/FGF1/ALPL(Fig 3b-c) and VCAN/CTNNB1/FGF1(Fig5) was statistically significant compared with the normal group. Furthermore, gene expression of NCAM/VCAN (Fig. 1D) seems not very significant between 3D and 2D cultures. Thus, the 3D models may be softly superior to the 2D models in this manuscript.
Response 1: We thank the reviewer for this comment. In figure 1D, the gene expression of VCAN was 4-time fold-change compared with the 2D. As for NCAM, it was a 2.5-fold change. According to statical analysis, these data were significant which shows that 3D system affected these genes more than 2D. As for Fig 5, TCQA upregulated all markers in 2D and 3D systems, but the effect was more significant in the 3D systems as shown in the revised version of the manuscript. In the supp table we can see the fold change in TCQA treated cells in 3D and 2D system where the fold change for VCAN in 2D system is 4-fold change but in case of 3D is 12-fold change.
- Comment 2: The authors should at least do the transplantation assay with replicates to verify the minoxidil or TCQA-treated cells have maintained or improved hair regenerative ability.
Response 2: We thank the reviewer for this comment. We also agree with the reviewer about the importance to perform a transplantation assay. But this study is a preliminary study, and we would like for future perceptive to further investigate this 3D system with TCQA and minoxidil in in vivo studies.
- Comment 3: It is unreasonable to state “To determine whether there is successful spheroid formation, we compared the consistency of spheroid size and shape formation using gene expression analysis” in lines 174-175.
Response 3: We do agree with reviewer and this sentence was removed in the revised version. We do agree that the upregulation of VCAN and NCAM correlate with ECM interaction and cell adhesion and do not reflect the size and the shape of the spheroids.
- Comment 4: The word minoxidil might be missing in the sentence “The cells were cultured in 3D and 2D system and treated with 0 and 0.1 μM for 48 h” in line 209.
Response 4: We do apologize for this careless mistake. The word minoxidil was added in the revised version.
- Comment 5: Please increase the font in the figures to make it more easily to be visualized.
Response 5: Thank you for this suggestion. The Font size was increased in all photos.
- Comment 6: Language should be improved. Many sentences are hard to follow.
Response 6: Thank you for your comment. The English was revised in all manuscript
Reviewer 3 Report
the authors did not considered my only comment. There is no answer to reviewer's comments and the modified manuscript is difficult to follow with track changes
Author Response
Response: We thank the reviewer for his comment, and we do apologize for the carelessness in the answer. We do understand the importance of his suggestions to further improve the quality of this paper. The changes were made in the discussion part of the revised manuscript in green highlight (as the track changes were difficult to follow), where we discussed the other current models of 3D culture and their application for ECM interactions. The changes regarding these comments were seen from lines 469-481, 515-517, and 535-541. Other changes regarding the English editing and general comments are highlighted in yellow.
Reviewer 4 Report
The authors have fully addressed my concerns.
Author Response
We thank the reviewer for all his suggestions and comments that helped to further improve the paper.
This manuscript is a resubmission of an earlier submission. The following is a list of the peer review reports and author responses from that submission.
Round 1
Reviewer 1 Report
An interesting article trying to demonstrate that the partial reprogramming of hair follicles dermal papilla cells by spheroid formation with the addition of hair growth stimulating compounds can restore inductive capacities of dermal papilla cells. Only minor queries:
In the statistical analysis subparagraph, please add the program you sued to calculate significance, its version, maker, and location.
line 77 you should add: "Various treatments have been proposed to manage alopecias, with food supplements like curcumin and innovative treatments such as blue light, with variable results" and cite : doi: 10.1111/dth.12842. and doi: 10.1007/s10103-021-03327-9.
line 460-462 is not written correctly...please modify.
Good Luck!
Reviewer 2 Report
Dermal papilla cells (DPc), usually grown in 2D culture, are widely used as an in vitro model to study hair growth. However, 2D-cultured DPc did not show an efficient therapeutic effect on hair follicle (HF) regeneration and growth. In the article titled “Surface Tension Guided Hanging-Drop: Producing Controllable 3D Spheroid of High-Passaged Human Dermal Papilla Cells and Forming Inductive Microtissues for Hair-Follicle Regeneration” published in “ACS Appl Mater Interfaces 2016 Mar 09:8(9) ” (DOI: 10.1021/acsami.6b00202), Lin et al showed that DPc grown in 3D hanging spheroids was more similarities to primary human DPc.
In the manuscript titled “Establishment of 3D Spheroid Human Dermal Papilla Cells as an Effective Model for Hair Growth Screening Compounds Compared with the 2D System: An Example of Minoxidil and 3,4,5-Tri-O-caffeoylquinic acid (TCQA)”, Bejaoui et al. compared the expression of signalling molecules and markers of HFDPC commercial cell line in untreated 3D and 2D cell cultures. Furthermore, Bejaoui et al. compared the expression of signalling molecules and metabolism-associated proteins of HFDPC in minoxidil and 3,4,5-tri-O-caffeoylquinic acid (TCQA) treated 3D and 2D cell cultures. Bejaoui et al. showed minoxidil or TCQA treatment to 3D cultured HFDPCs highly affected hair growth-associated pathway and markers, and cell cycle arrest signalling compared with the 2D model. This is generally a good preliminary study with a comparison of gene expression between 2D or 3D cultured cells with or without minoxidil or TCQA treatment. Minor concerns as follows.
- In both 3D and 2D models, gene expression of CTNNB1/FGF1/ALPL(Fig 3b-c) and VCAN/CTNNB1/FGF1(Fig5) was statistically significant compared with the normal group. Furthermore, gene expression of NCAM/VCAN (Fig. 1D) seems not very significant between 3D and 2D cultures. Thus, the 3D models may be softly superior to 2D models in this manuscript.
- The authors should at least do the transplantation assay with replicates to verify the minoxidil or TCQA-treated cells have maintained or improved hair regenerative ability.
- It is unreasonable to state “To determine whether there is successful spheroid formation, we compared the consistency of spheroid size and shape formation using gene expression analysis” in lines 174-175.
- The word minoxidil might be missing in the sentence “The cells were cultured in 3D and 2D system and treated with 0 and 0.1 μM for 48 h” in line 209.
- Please increase the font in the figures to make it more easily to be visualized.
6. Language should be improved. Many sentences are hard to follow.
Reviewer 3 Report
The authors present an interesting work comparing 2D vs. 3D spheroids culture for modelling hair follicle in-vitro. The study is very complete and rearch methods and results are clearly presenteed. To improve the quality of the paper I just suggest to discuss alternatives spheroids production methods, such as cell encapsulation in alginate and other matrix componets such as collagen. Indeed, this system can improve the relevance of the model providing cells with an ECM like 3D substrates (see for example https://www.mdpi.com/2073-4409/10/6/1458)
Reviewer 4 Report
The manuscript 1623600 by Bejaoui et al. is about growing human dermal papilla cells as 3D spheroid cultures, instead of 2D cultures, to develop an in vitro cell system better suited to study hair follicle and to screen for compounds that could stimulate hair growth. I have decided, after reading twice the manuscript, that it does not have the necessary quality to go forward with publication. There are too many typos, the presentation of the data is poor, and the English grammar has to be drastically improved. Independently of whether results are interesting or not, the poor quality of the submission 1623600 clearly indicate that the authors did not put enough efforts in carefully reviewing and maturating their manuscript before submission. Authors should improve their manuscript before they resubmit it. I have written down some general comments that should be considered before resubmitting the revised version of the manuscript.
Title should be revised for grammar but also for accuracy. Cell culture is 3D, not cells. Screening of compounds not screening compounds. Tile is also too long.
You have to be clear in your affirmations. Examples: Line 19, it is unclear how 2D cell cultures of DPCs can have a therapeutic effect on HF regeneration. Line 66, 3D culture system cannot get insights. Line 174, how do you assess spheroid size and shape through gene expression analysis?
You have to be precise in your statements. Examples: Line 23, variability of what? Gene signature for what? which signalling molecules? Line 27, which validation? Line 43, which disturbances? Line 45, which dysregulation? Line 47, which irregularities? You are too vague and should give some examples and more details. Line 209: 0.1 µM of what? Line 212: Figure S2 is cited for figure S1.
You have to clearly exposed the objectives of your study at the end of the introduction. What is the scientific question that you want to answer? What do you propose to answer this question?
Materials and methods section lacks details. Examples: Line 104, what is the “appropriate medium”, what are the growth factors used to supplement culture medium? Line 107, what is the cell density used to seed the cells? How do you passage the cells? What is the collagen concentration? What is room temperature? Line 127, what is an Isogen extraction? Line 130, how RNA integrity was assessed? Line 148, define gene names, define qPCR conditions?
Figure 1: indicate some few genes of interest in the volcano plot to make it interesting. Put some effort in the presentation of the data. What is the interest of panel C? wwhy only showing up-regulated genes? Panel E, how can you have signal for NCAM and VCAN on the same blot ?!?!? Legend of the y axis of the graph is definitely not band quantification. What are the units, etc…..
Figure 2: homogenize data presentation and highlight the important information. A bit boring as it is.
Figure 3: ALP or ALPL? You tested the significance of the differences between CTRL and Minoxidil but nether between 2D and 3D. However, you concluded in the text on a significant increase of gene expression in 3D cell cultures.
Figure 4: improve picture quality and include a scale for the size. Highlight important information. Where is panel E? Panel G, 3D control instead of 2D control for the blue circle.
Figure 5: VCAN in italic. Check gene name nomenclature throughout the text.
Discussion: Do not repeat the Results. There is also in the Discussion some literature that should in the Introduction to better understand the aim of this paper. You should also discuss gene expression more globally. Why a higher fold change in gene expression is a better result? Why having more differentially expressed genes is a better result? I do not say that it is wrong but you have to convince me and bring some supporting literature data.
Finally, a scheme that would present all this expression data in a more informative way (gene tables and ontology graphs are boring) would be greatly appreciated.